# Mechanical Behavior of Polyurethane Insulation of CRT Leads in Cardiac Implantable Electronic Devices: A Comparative Analysis of In Vivo Exposure and Residual Properties

**DOI:** 10.3390/bioengineering11020156

**Published:** 2024-02-04

**Authors:** Anmar Salih, Tarun Goswami

**Affiliations:** 1Department of Biomedical, Industrial and Human Factors Engineering, Wright State University, Dayton, OH 45435, USA; salih.7@wright.edu; 2Department of Orthopedic Surgery, Sports Medicine and Rehabilitation, Miami Valley Hospital, Dayton, OH 45409, USA

**Keywords:** polyurethane, lead insulation, left ventricle leads, failure analysis, in vivo environment

## Abstract

Left ventricle leads are designed for the purpose of long-term pacing in the left ventricle. This study investigated the leads that use polyurethane as an outer insulator and SI-polyimide as an inner insulator. Polyurethane is commonly used for the outer insulation of cardiac leads due to its flexibility and biocompatibility. SI-polyimide (SI-PI) is a high-performance material known for its electrical insulation properties and is used for the inner insulation to maintain the integrity of the electrical pathways within the lead. Ten leads were received from the Wright State University Anatomical Gift Program. The duration of in vivo implantation varied for each lead, from less than a month to 108 months, with an average in vivo duration of 41 ± 31 months. We used the Test Resources Q series system for conducting our tests, as well as samples prepared to ensure compliance with the ASTM Standard D 1708-02a and the ASTM Standard D 412-06a. During the test, the load was applied to the intact lead. Before conducting individual tests, each lead was carefully inspected for surface defects. After conducting the tests, the load to failure, percentage of elongation, percentage of elongation at 5 N, ultimate tensile strength, and modulus of elasticity were calculated. There was no significant difference in load to failure, the percentage of elongation to failure, ultimate tensile strength, and modulus of elasticity (*p*-value = 0.82, *p*-value = 0.62, *p*-value = 0.82, and *p*-value = 0.12), respectively, when compared to in vivo exposure time. On the other hand, the percentage of elongation at 5 N force showed a significant difference (*p*-value = 0.0066) after 60 months in an in vivo environment. As the duration of in vivo exposure increased, the load to failure, percentage of elongation, ultimate tensile strength, and modulus of elasticity decreased insignificantly. The residual properties of these left ventricle leads remained relatively stable after 108 months of in vivo exposure duration, with no statistically significant degradation or changes in performance.

## 1. Introduction

Attain Performa 4298, Attain Performa Straight 4398, and Attain Performa S 4598 leads are quadripolar electrode leads, are 78 cm or 88 cm in length, 5.3 F (1.75 mm) in diameter, and are designed for the purpose of long-term pacing and sensing in the left ventricle [1]. These leads are serving the same purpose; however, they differ in shape, as shown in Figure 1. These leads use polyurethane (55D) as an outer insulator and SI-polyimide as an inner insulator [2]. These leads received FDA approval in 2014, and since then, there have been more than 218,000 leads implanted in the United States alone [1].

Polyurethane is a versatile polymer that has been in use since the 1930s. It was first developed in Germany by Dr. Otto Bayer and his colleagues at the chemical company Bayer AG. Over the years, polyurethane has been used for various industries, including the medical field. It has been used as an insulating material for medical devices, such as cardiac leads, catheters, and implantable devices, because of its biocompatibility and electrical insulation properties [3]. Polyurethane is a versatile polymer that is formed by the reaction of polyols with diisocyanates [4]. The design considerations may involve selecting specific types of polyols and diisocyanates, as well as incorporating additives for desired properties [4]. Polyurethane exhibits good mechanical strength and high tensile strength, it can withstand bending without breaking easily, and it exhibits toughness and impact resistance [5]. Polyurethane materials can have inherent lubricious properties, providing a low coefficient of friction. The lubricity can contribute to the wear resistance of polyurethane components [6,7]. SI-polyimides are a class of high-performance polymers with exceptional thermal stability and electrical insulating properties. SI-polyimides, in particular, are known for their superior performance in high-temperature and harsh environments. Polyimides were first synthesized in the mid-20th century, with their development attributed to Joseph Campbell [8]. The design of SI-Polyimides involves incorporating chemical structures that allow the material to undergo controlled depolymerization and subsequent repolymerization, enabling self-repair under certain conditions. SI-Polyimides are known for their high-temperature stability, high tensile strength, relatively stiff materials, and creep resistance [8]. Polyimides generally have good lubricity due to their low coefficient of friction. However, the lubricious properties may vary based on the specific formulation of the polyimide [9].

SI-polyimides have found applications in various fields, including the medical device industry. They are used as insulation materials in implantable medical devices like leads and catheters due to their ability to maintain electrical insulation properties even in demanding conditions [8]. Polyurethane and SI-polyimide (SI-PI) are chosen for their biocompatibility, electrical insulation properties, and durability when implanted in the human body. Polyurethane is commonly used for the outer insulation of cardiac leads due to its flexibility and biocompatibility [9]. SI-polyimide (SI-PI) is a high-performance material known for its electrical insulation properties and is used for the inner insulation to maintain the integrity of the electrical pathways within the lead. The selection of these materials is based on thorough testing to ensure their safety and effectiveness in a medical device context, where they will be in contact with tissues and fluids for a long period [9]. This combination of inner and outer insulation materials helps provide reliable and safe electrical conduction within the CRT lead while minimizing the risk of complications associated with the lead’s implantation.

Studying the residual properties of leads proved to be the most challenging aspect of this research. Only a limited number of studies explored how these properties degrade within an in vivo environment. For example, Wilkoff et al. [10] examined three different types of insulations—Optim, P55D, and silicone elastomer. These leads were divided into three categories based on the duration of in vivo exposure (zero years, 2–3 years, and 4–5 years). Subsequently, a tensile test was conducted to measure the maximum load and extension of these leads. The results revealed that the molecular weight of Optim decreased by 20% after 2–3 years and then remained constant during the subsequent 4–5 years. In contrast, the tensile strength decreased by 25% after 2–3 years and then stabilized over the next 4–5 years. Interestingly, the elongation of Optim remained unchanged throughout the study period. Notably, the molecular weight of polyurethane exhibited no alterations during this timeframe. Silicone, on the other hand, demonstrated significant biostability when compared to polyurethane and Optim [11]. Other studies either investigated the mechanical damage of the lead and how this damage could affect the functionality of the cardiac implantable electronic devices, or investigated the polyurethane insulation in other medical devices, like neurostimulators [12,13].

## 2. Materials and Methods

We utilized ten left ventricle pacing leads in this study. These leads are passive fixation, quadripolar, coaxial design type with lengths of 78 and 88 cm. These leads use polyurethane as an outer insulator and SI-polyimide as an inner insulator [2] and are manufactured by Medtronic, Minneapolis, MN, USA. The testing was conducted using the Test Resources Q series system, which applies specific displacements or loads to various samples and measures the corresponding load or displacement. The XY software Manual v. 0.9.0 on the connected computer acquires both load and displacement values, along with sampling points, ensuring precise data capture during the testing process. The in vivo exposure varied for each lead, ranging from less than a month to 108 months, with an average in vivo duration of 41 ± 31 months. Figure 2 illustrates the test procedure, the testing fixture, and a cross-section of a sample under a microscope, revealing the coils and insulators. Our tests adhered to the ASTM Standard D 1708-02a [14] and the ASTM Standard D 412-06a [15].

All tested leads had a fixed sample length of 38 mm, with 8 mm of the length held by the grips and 22 mm positioned between them. We tested the intact left ventricle leads, including both the outer insulator (polyurethane) and the inner insulator (SI-polyimide) with the coil inside the insulation. This approach aimed to simulate the real-life conditions in which the leads would be functioning within the human body, ensuring a comprehensive evaluation of the mechanical behavior of the entire lead assembly, and, to prevent slippage, the leads were securely affixed in the grips using sandpaper. During the tensile test, specific loads were applied to the samples, and the corresponding displacement was measured. The tensile test was conducted at least two to four times, and the results were averaged. In addition, the diameter of each specimen was measured at three locations, and the average diameter was measured to be 1.75 mm. A standardized gauge of 22 mm length was employed for all specimens, with 8 mm inside the upper and lower grips. Furthermore, all leads were examined under an optical microscope to assess any damage, both before and after the tests. The tensile test was conducted at a rate of 1 mm/second, with close observation of the lead’s extension. Following the separation of the lead’s insulation, we calculated key parameters, including load to failure, elongation to failure, the percentage of elongation at 5 N, ultimate tensile strength, and the modulus of elasticity. Finally, we compared this data, considering the in vivo exposure duration in years.

## 3. Results

### 3.1. Load to Failure

The dataset provides insights into the mechanical performance of the leads; the maximum load to failure recorded was 62.8 N, the minimum load to failure was 43.7 N, the mean load to failure was 53.4 N, and the median load to failure was 54.2 N. Upon conducting a statistical analysis, it was determined that there was no statistically significant difference observed with respect to the in vivo exposure time (*p*-value = 0.82). Specifically, the load to failure was 54.4 ± 8.2 N at less than 1 month of in vivo exposure, and this value exhibited a minor decrease to 53.5 ± 4.5 N at 24 months. Subsequently, it demonstrated fluctuations as the exposure time increased, stabilizing at 53.4 ± 2.9 N in 38 months. It is notable that the load-to-failure value exhibited a gradual decrease as the in vivo exposure increased to 108 months, as shown in Figure 3.

### 3.2. Elongation to Failure

The data collected showed that there was no significant difference in maximum elongation with respect to the in vivo exposure time (*p*-value = 0.62). The maximum elongation to break recorded was 284%, the minimum elongation to break was 192.9%, the mean elongation to break was 251.9 ± 22.9%, and the median load to failure was 259.3%. The maximum elongation at less than one month of in vivo exposure was 256.9 ± 27.3%. This value fluctuated down to 249.5 ± 26.1 after 24 months, rising to 266.7 ± 4.7% at 31 months of in vivo exposure. The value of the percentage of elongation declined to 224.4 ± 27.2% after 64 months of in vivo exposure, as shown in Figure 4. The overall value continued to decrease with in vivo exposure, but with no significant difference up to 108 months.

### 3.3. Percentage of Elongation at 5 N Force

This study examined the percentage of elongation under a 5 N force, considering previous research indicating that the maximum in vivo load applied to the lead falls within the 5 N range [16]. Moreover, the traction forces identified in our investigation align consistently with the traction forces specified in the existing EN standards. In accordance with EN standard 45502-2-1, a lead must withstand a traction force of a minimum of 5 N for at least 1 min to meet the criteria for market approval [17]. The percentage of elongation observed at 5 N force was found to be consistent with the percentage of elongation during load to failure and the resulting percentage of elongation after the tests. The mean percentage of elongation at 5 N was 5.9 ± 0.79%, with a maximum of 7.33%, minimum of 4.32%, and median of 5.94%. Statistical analysis showed that there was a significant difference with respect to in vivo exposure time (*p*-value = 0.0066). There were two groups to compare in this statistical test. Group one was the in vivo exposure of less than 1, 25, and 108 months, and group two was the rest of the in vivo exposure times, as shown in Figure 5. The two groups had significant differences between them; group one was significantly higher than group two, and that fluctuation also can be seen on the percentage of elongation, as well. Although there was a fluctuation, it can be seen clearly that the 5 N force elongation is incrementing with the increase in the in vivo exposure time, as illustrated in Figure 6.

### 3.4. Ultimate Tensile Strength

This study focused on assessing the ultimate tensile strength of the Attain Performa 4298, 4398, and 4598 left ventricle leads. The calculated mean ultimate tensile strength was 22.1 ± 1.96 MPa, with a maximum value of 26.01 MPa, a minimum value of 18.15 MPa, and a median of 22.5 MPa. There were no statistically significant differences associated with the in vivo exposure time (*p*-value = 0.82). However, it is worth noting that there was a noticeable decline in the ultimate tensile strength as the duration of in vivo exposure increased, as illustrated in Figure 7. The initial ultimate tensile strength at less than 1 month of in vivo environment exposure was 22.6 ± 3.4 MPa, showing a slight increase to 23.4 ± 1.3 MPa at 25 months of in vivo exposure. Subsequently, the ultimate tensile strength began to decrease, reaching 20.4 ± 1.8 MPa at 64 months of in vivo exposure and ultimately measuring 21.8 ± 3.2 MPa after 108 months of in vivo environmental exposure. The data reflect a trend of declining ultimate tensile strength with prolonged exposure to the in vivo environment, even though statistical significance was not observed.

### 3.5. Modulus of Elasticity

This research investigated the modulus of elasticity of polyurethane insulation leads, and we found that there was no statistical difference when compared with in vivo exposure times (with a *p*-value = 0.12). The mean modulus of elasticity was found to be 20.8 ± 2.3 MPa. The modulus of elasticity ranged from a minimum of 15.9 MPa to a maximum of 25.4 MPa, with a median value of 21.24 MPa. The initial modulus of elasticity was 19.3 ± 1.3 MPa when measured at less than one month of in vivo exposure, as shown in Figure 8. However, this value showed an increase to 23.1 ± 1.9 MPa after 24 months of exposure and was stable at 23.6 MPa after 60 months of in vivo exposure. With longer exposure, the modulus of elasticity began to decline, reaching a value of 20.2 ± 1.1 MPa after 108 months of in vivo exposure.

All in vivo residual properties are tabulated in Table 1 below, and load versus extension curves for all in vivo exposure times are summarized in Figure 9.

## 4. Discussion

The insulation with the coil was tested as a whole unit, meaning the whole left ventricle lead, including both the outer insulator (polyurethane) and the inner insulator (SI-polyimide) with the coil inside the insulation. This information may help prevent unintended insulator breaches, which can lead to tissue damage and the inability to deliver therapy. In vivo conditions, including temperature, moisture, and exposure to chemicals, can play a significant role in material degradation. The pliable components within polyurethanes, specifically those composed of polyethers, are prone to oxidative deterioration, encompassing environmental stress cracking (ESC) and metal ion oxidation (MIO) [18,19,20]. Variations in these environmental factors over time might explain fluctuations in the results, patient condition, and percentage of usage of the lead. Temperature can have a profound impact on the degradation of materials within a living organism.

Polyetherurethane (PEU) demonstrates remarkable resistance to substantial biological degradation caused by substances produced by phagocytic cells, such as cationic proteins, proteases, superoxide, and hydrogen peroxide [21]. However, a significant portion of neutrophils effectively transforms considerable quantities of hydrogen peroxide into the more oxidative agent, hypochlorous acid [22]. This process is believed to be a probable contributor to the in vivo degradation of PEU [23]. In left ventricle leads equipped with PEU insulation, alongside the deterioration caused by environmental stress cracking occurring in direct contact with tissue, there was an observed occurrence of metal ion oxidation (MIO). MIO represents a mechanism of PEU degradation that arises when in direct contact with metal ions originating from conductor coils. These metal ions come into contact with hydrogen peroxide, a result of inflammatory cells, which penetrates through the outer insulation. MIO is a concern for both the inner surface of the outer insulation and, in general, for both the outer and inner surfaces of the inner insulation. This phenomenon impacts various layers of insulation, making it a critical consideration in the performance and safety of these medical devices [24]. Polyether polyurethane elastomers, despite not being biostable, remain in use due to their exceptional mechanical characteristics and biocompatibility. Their continued application is recognized as being due to advancements in comprehending and managing the processes by which they degrade. These developments have allowed for more effective control over their degradation mechanisms, ensuring their viability for various applications [24]. Clinical research has investigated the factors that pose risks for lead damage, examined how lead failures manifest, and put forward strategies for their management. In particular, transvenous endocardial leads that feature PU55D insulation exhibit a heightened susceptibility to damage. It is also important to note that the copolymer of polyurethane and silicone are not resilient to thermal damage caused by cautery-cut modes [25].

Polyurethane mechanical properties are highly temperature dependent [26]. At lower temperatures, polyurethane tends to become stiffer and less flexible. This is because the polymer chains are held together more tightly due to reduced thermal energy. As the temperature increases, the material becomes more elastic and pliable, which can be advantageous for applications requiring flexibility and impact resistance [26]. The tensile strength of polyurethane often decreases at higher temperatures. This is because the elevated temperature weakens the intermolecular forces that hold the polymer together [27]. Polyurethane is known for its excellent impact resistance, and it exhibits viscoelastic behavior [27]. Both of these characteristics can be influenced by the temperature within an in vivo environment, which brings us to the findings presented in Table 2. It is notable that, as the duration of in vivo exposure increased, four out of the five residual properties in this study exhibited a decline.

Over time, mechanical wear, tear, and creep can occur due to constant movement and physiological factors in an in vivo environment. This wear could lead to a gradual reduction in mechanical properties [28]. Polyurethane is a versatile polymer known for its durability and elasticity, but when it undergoes wear and tear, its residual properties can be affected by abrasion, surface wear, fracture toughness, the coefficient of friction, and finally, fatigue resistance. To mitigate the effects of wear on polyurethane mechanical properties, strategies such as incorporating wear-resistant additives, optimizing surface treatments, and selecting appropriate formulations can be employed [28].

Our research presented a mathematical prediction model for load to failure, the percentage of elongation to failure, percentage of elongation at 5 N, ultimate tensile stress, and modulus of elasticity. All these equations are summarized in Table 2 below.

Each equation represents the prediction of how this characteristic would behave during its in vivo exposure time, where τ represents the in vivo exposure time. All models showed a decrease in their properties while inside the body, except for the percentage of elongation at 5 N force. This may be attributed to the gradual degradation of the polyurethane insulation material over time. As the polyurethane insulation material degrades, it may become less stiff and more prone to elongation under lower loads, leading to the observed increase in the percentage of elongation at 5 N force with longer exposure times [29,30].

A residual versus predicted plot was generated to assess the goodness of fit of the model, as shown in Figure 10 and Figure 11. Residuals are the differences between the observed values and the predicted values from the regression model. Ideally, the residuals must be randomly scattered around zero. All graphs and plots generated are randomly distributed around zero, with scattered patterns. The spread of residuals should be roughly constant across all levels of the predicted values. The spread of residuals widens or narrows as the predicted values increase. Data contain outliers, and these outliers can significantly influence the model. All data from the residual versus predicted plot as shown in Figure 10 show a random scattering of points around zero, which suggests that the model is a good fit for the data. An actual versus predicted plot typically compares observed or measured values (actual) with values predicted by a model. This means that the predicted values were within the band of actual values. Then, when we examined the distribution of the points, it suggested a good fit between the actual and predicted values; no deviation was found in the data. A tight distribution means good precision, while a scattered pattern suggests variability or inconsistency in predictions. From what we had, a Monte Carlo simulation for the data was generated to check the probability of the distribution.

After conducting a Monte Carlo simulation on load-to-failure, percentage of elongation to failure, 5 N percentage of elongation, ultimate tensile strength, and modulus of elasticity failure probability values, we applied the most conservative failure distribution to identify the point where the likelihood of failure was highest. In our analysis, the X column denotes the probability of failure for different mechanical properties, ranging from 0 to 1, reflecting the varying likelihood of failure. The corresponding Y column represents the mechanical property values associated with these failure probabilities.

The dataset of load to failure illustrates a clear trend, as shown in Figure 12A; as the failure probability increases from 0.0 to 1.0, the corresponding load-to-failure quantile also rises. This consistent relationship underscores the link between failure probability and load-to-failure values. Specifically, at a 0% failure probability, the load-to-failure quantile was 36.4 N. However, as the likelihood of failure increased, the load-to-failure quantile progressively climbed, peaking at 57 N when the failure probability reached 1.0. This progression indicates a positive correlation between failure probability and the load-to-failure quantile. As the chances of failure grow, so does the load-to-failure quantile, representing the associated time or population where a certain percentage of failures occur. The data, as shown in Figure 12, reveal a systematic connection between the likelihood of failure and the corresponding load-to-failure quantile, providing a quantitative understanding of the system’s reliability across various failure scenarios.

The quantification of elongation to failure underwent a Monte Carlo simulation to enhance precision. As the failure probability ascended from 0.0 to 1.0, a corresponding augmentation in the percentage of elongation was observed. This data explain a systematic and incremental pattern, highlighting a proportional relationship between failure probability and the percentage of elongation. Specifically, with a failure probability of 0.0, the percentage of elongation stood at 174.7%. Subsequently, this elongation metric progressively increased with escalating failure probabilities, closing in a peak value of 268.2% at a failure probability of 1.0, as shown in Figure 12B. The observed progression of values strongly suggests a positive correlation between failure probability and the percentage of elongation. This implies that, as the likelihood of failure intensifies, the material tends to exhibit greater elongation before reaching its breaking point. This nuanced understanding of the material’s behavior under varying failure scenarios is invaluable for applications in materials testing and engineering analyses.

The Monte Carlo simulation was applied to predict the failure probability of Polyurethane at 5 N elongation, focusing on the percentage of elongation. As the failure probability escalated from 0 to 1, there was a corresponding increase in the percentage of elongation at the 5 N quantile. The observed values followed a continuous and incremental pattern, highlighting a positive correlation between failure probability and the percentage of elongation at 5 N, as illustrated in Figure 12C. At a failure probability of 0, the percentage of elongation at the 5 N quantile was 2.8%. This metric gradually rose with increasing failure probability, reaching 6.5% when the failure probability was 1. The progression in the data suggests that as the likelihood of failure increases, the Polyurethane material undergoes a more substantial percentage of elongation before reaching failure at a 5 N load. This information, depicted in Figure 12C, contributes valuable insights into the material’s behavior under varying stress conditions and aids in assessing its ductility and performance characteristics.

We explored another mechanical aspect, using a Monte Carlo simulation to gauge the likelihood of failure regarding ultimate tensile strength, as illustrated in Figure 12D. Ultimate tensile strength is a critical material property, defining the maximum stress a material can endure before breaking. The dataset reveals a clear trend; as the failure probability climbs from 0.0 to 1.0, the corresponding ultimate tensile strength quantile also climbs. The values unfold a steady and incremental progression, hinting at a proportional connection between failure probability and ultimate tensile strength quantile. At a failure probability of 0.0, the ultimate tensile strength quantile stood at 15.2 MPa. This quantile steadily ascended with growing failure probabilities, peaking at 23.6 MPa when the failure probability reached 1.0. The graphical representation illustrates that any modulus of elasticity value below 20.4 MPa carries an almost negligible probability of failure. This analysis provides valuable insights into the interplay between the modulus of elasticity and failure probability, revealing a critical threshold, below which the risk of failure is substantially reduced. The design considerations for the utilization of the leads are based on a 0.5 failure probability, limiting the ultimate tensile strength to be nearly 22.5 MPa. This consistent trend implies a positive correlation between failure probability and the ultimate tensile strength quantile. Essentially, as the likelihood of failure increases, the material tends to manifest higher ultimate tensile strength before reaching its breaking point. These findings contribute significant insights into the material’s behavior under various stress conditions, offering valuable information for assessing its robustness and performance characteristics.

## 5. Conclusions

This study revealed a notable degree of stability in Attain Performa 4298, Attain Performa Straight 4398, and Attain Performa S 4598 left ventricle leads’ residual properties across a wide range of in vivo exposure durations. There was no significant difference in load to failure, the percentage of elongation to failure, ultimate tensile strength, and modulus of elasticity (*p*-value = 0.82, *p*-value = 0.62, *p*-value = 0.82, and *p*-value = 0.12), respectively, when compared to in vivo exposure time. On the other hand, the percentage of elongation at 5 N force increased slightly but significantly (*p*-value = 0.0066) after 108 months in an in vivo environment; however, no failure is to be expected from elongation up to 5 N. The absence of statistically significant degradation or alterations in performance suggests a robust and reliable performance of these leads under varying physiological conditions. As the duration of in vivo exposure increased, the load to failure, percentage of elongation, ultimate tensile strength, and modulus of elasticity decreased insignificantly. Even though in vivo exposure introduces a multitude of factors, including temperature fluctuations, moisture levels, and exposure to diverse chemicals within the body, all of which can collectively contribute to material degradation, these leads remained stable up to 108 months in an in vivo environment. This investigation highlights the critical role of polyurethane insulation in the mechanical behavior of these leads. Ensuring the consistency and high quality of polyurethane insulation material emerges as a paramount consideration, especially in scenarios where the mechanical performance of the material is of utmost importance, such as in Cardiac Implantable Electronic Device (CIED) leads.

## 6. Limitation

This investigation encountered some limitations. The testing conditions and procedures used in the experiment may impact the results. Variations in testing equipment, techniques, or protocols could introduce variability. The lack of a wide range of in vivo duration was one of the limitations of this research. We could have more insight if we had more leads with a longer implant duration.

## Figures and Tables

**Figure 1 bioengineering-11-00156-f001:**
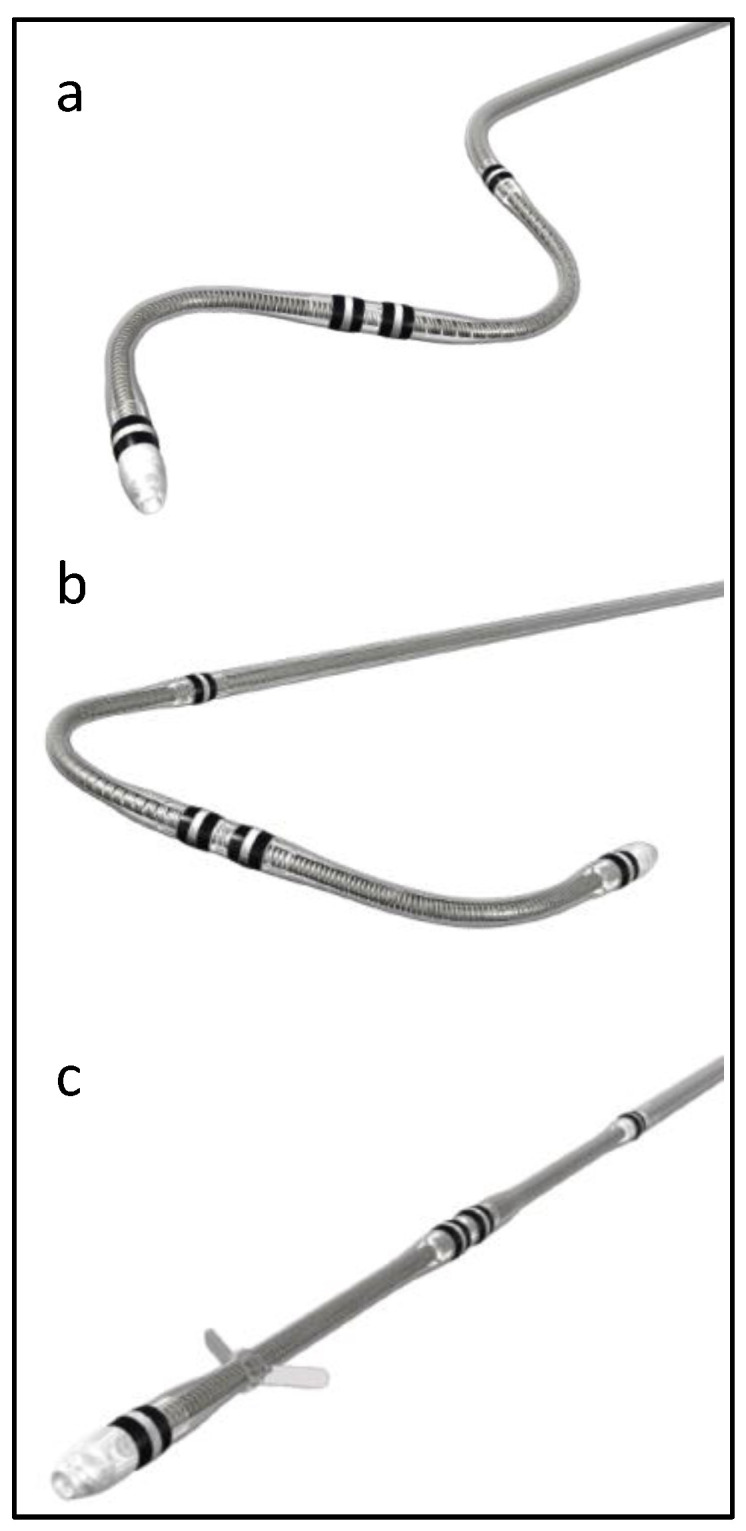
Attain Performa CRT lead shapes, (**a**) Attain Performa S 4598, (**b**) Attain Performa 4298, and (**c**) Attain Performa Straight 4398.

**Figure 2 bioengineering-11-00156-f002:**
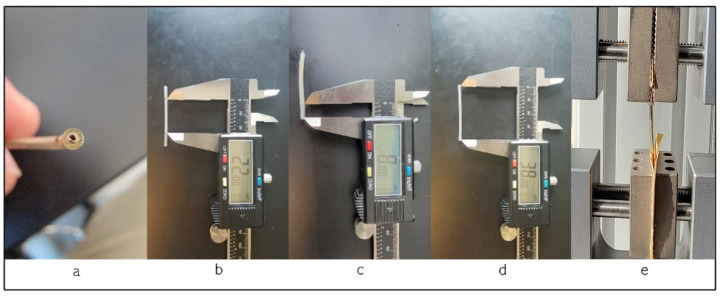
(**a**) Cross-section of the lead; (**b**) length of the specimen between the grips, showing the length of 22 mm; (**c**) length of the lead in the grip, 8 mm; (**d**) the whole length of the specimen; (**e**) during the test procedure.

**Figure 3 bioengineering-11-00156-f003:**
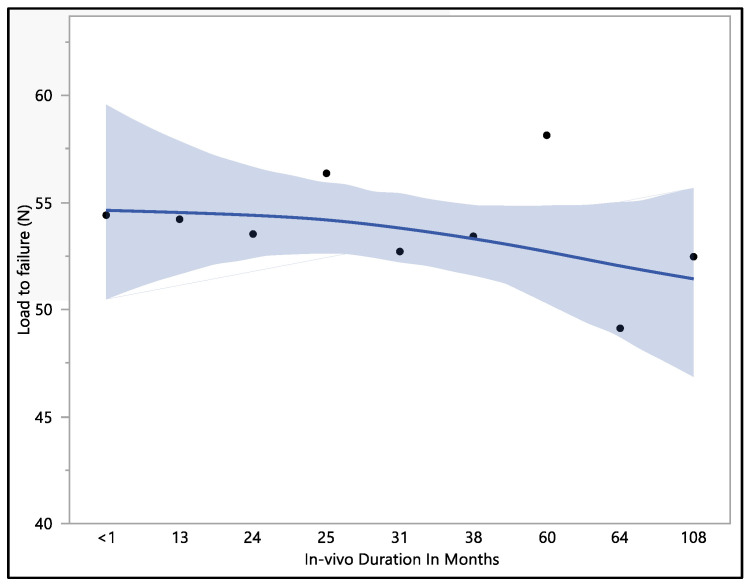
Representative mean load to failure vs. in vivo months plot of Attain Performa 4298, Attain Performa Straight 4398, and Attain Performa S 4598 left ventricle leads.

**Figure 4 bioengineering-11-00156-f004:**
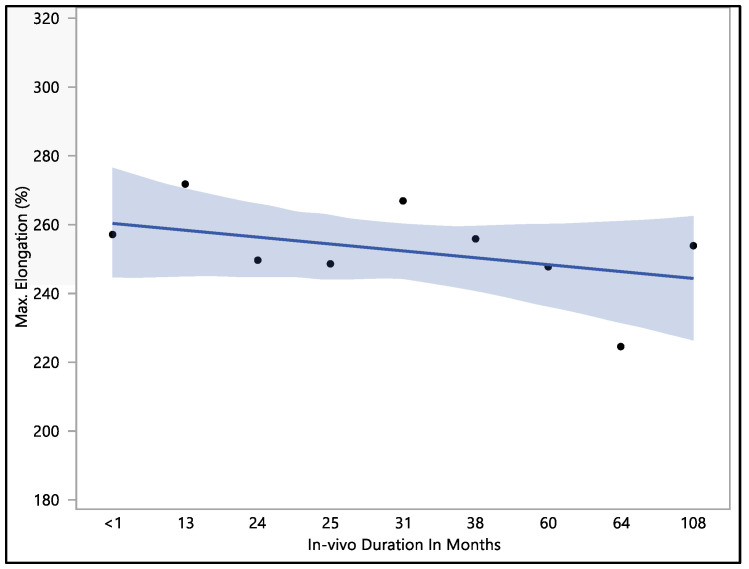
Representative plot of mean elongation to break vs. in vivo months of Attain Performa 4298, Attain Performa Straight 4398, and Attain Performa S 4598 left ventricle leads.

**Figure 5 bioengineering-11-00156-f005:**
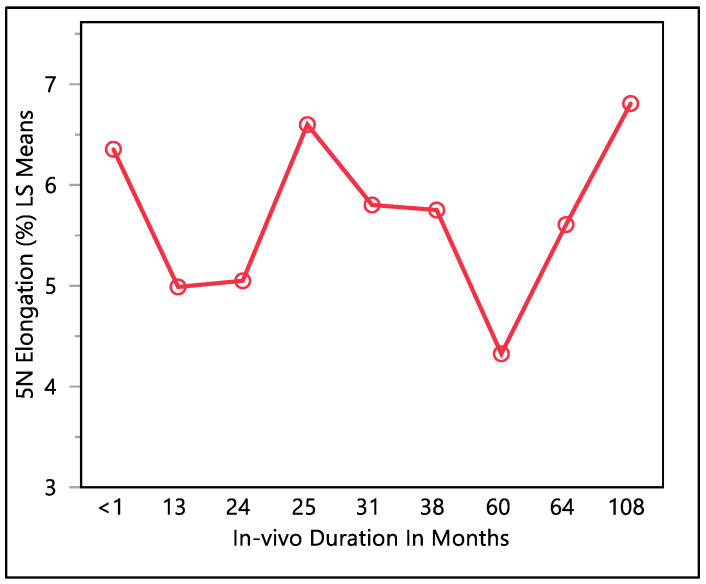
LMS squares means plot.

**Figure 6 bioengineering-11-00156-f006:**
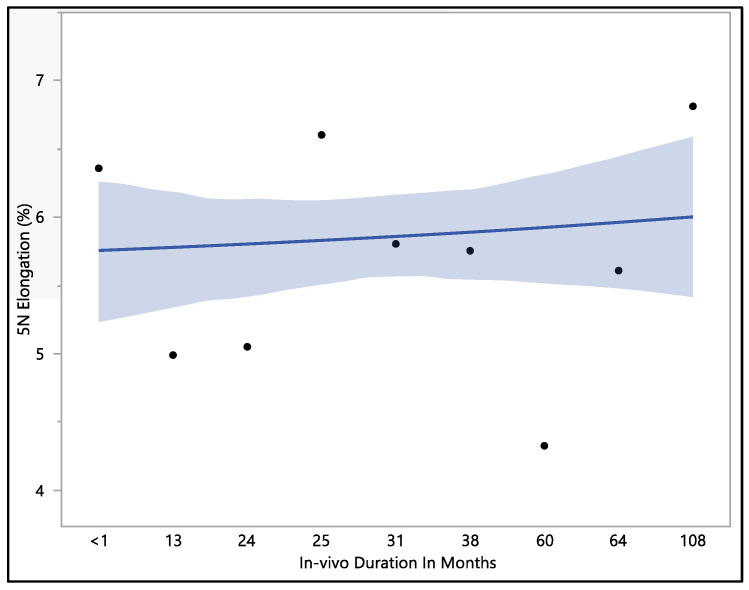
Representative plot of mean elongation at 5 N vs. in vivo months of Attain Performa 4298, Attain Performa Straight 4398, and Attain Performa S 4598 left ventricle leads.

**Figure 7 bioengineering-11-00156-f007:**
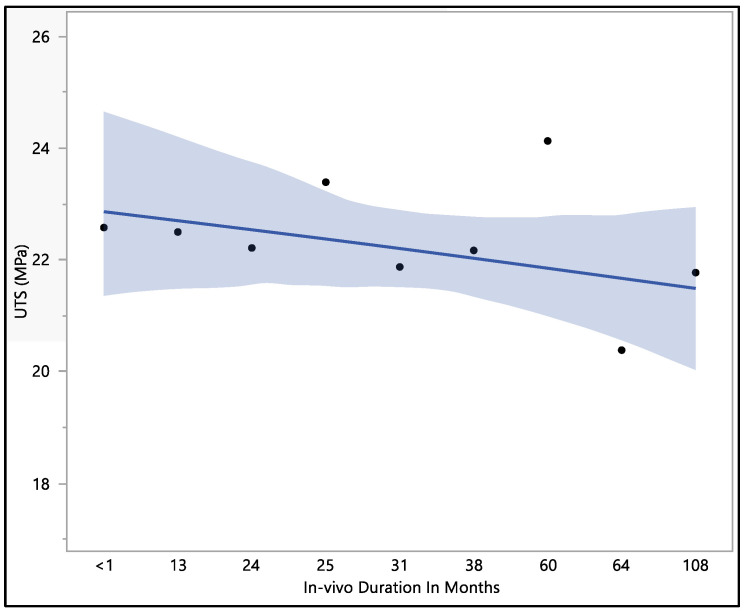
Representative plot of mean of ultimate tensile strength vs. in vivo months of Attain Performa 4298, Attain Performa Straight 4398, and Attain Performa S 4598 left ventricle leads.

**Figure 8 bioengineering-11-00156-f008:**
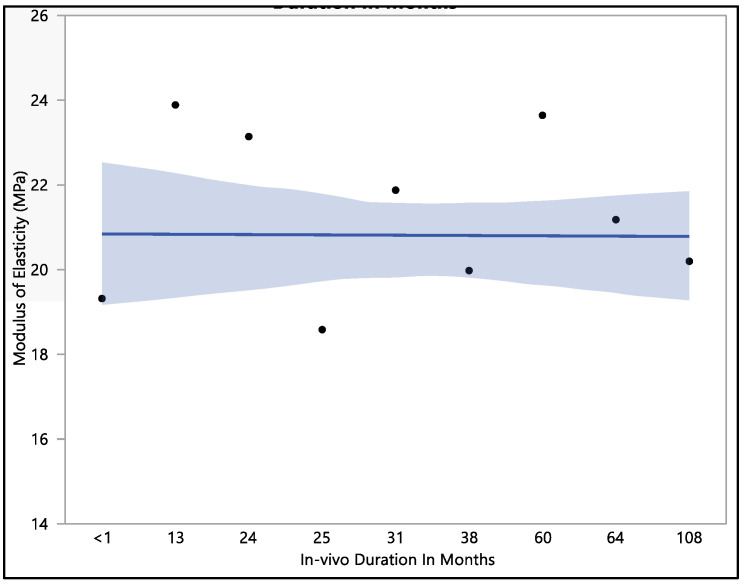
Representative plot of mean of modulus of elasticity vs in vivo months of Attain Performa 4298, Attain Performa Straight 4398, and Attain Performa S 4598 left ventricle leads.

**Figure 9 bioengineering-11-00156-f009:**
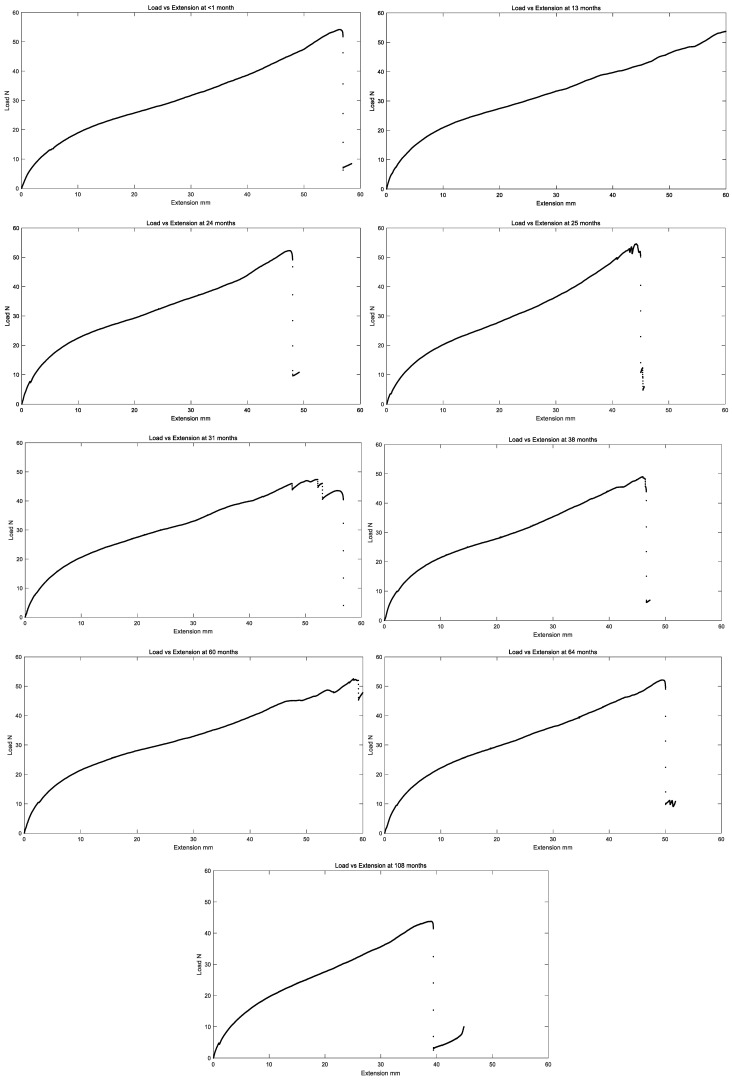
Representative plot of load vs. extension of the left ventricle polyurethane leads for different in vivo implantation durations.

**Figure 10 bioengineering-11-00156-f010:**
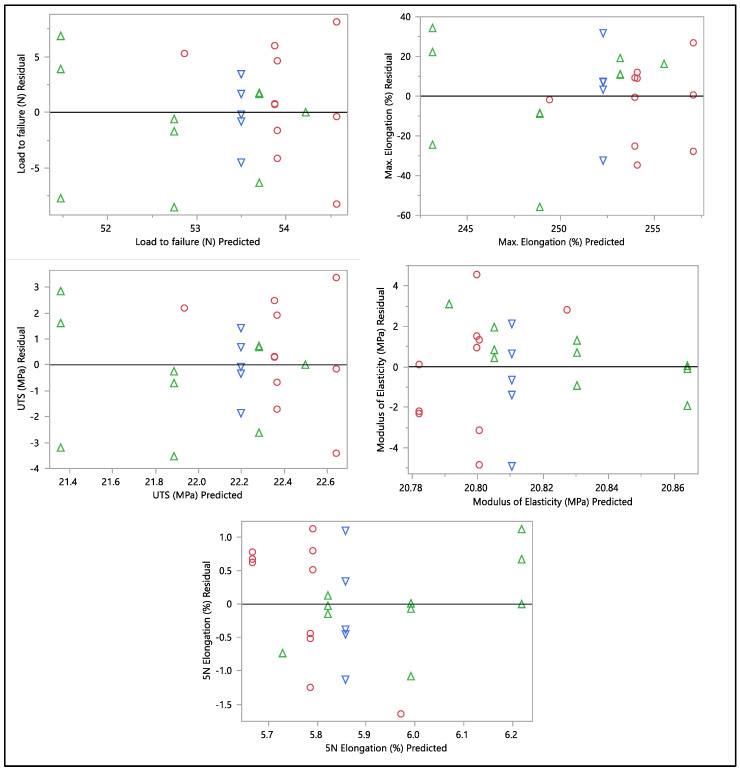
Residual vs. predicted plots of load to failure, percentage of elongation to failure, 5 N percentage of elongation, ultimate tensile strength, and modulus of elasticity. Different colors mean different in-vivo duration exposure.

**Figure 11 bioengineering-11-00156-f011:**
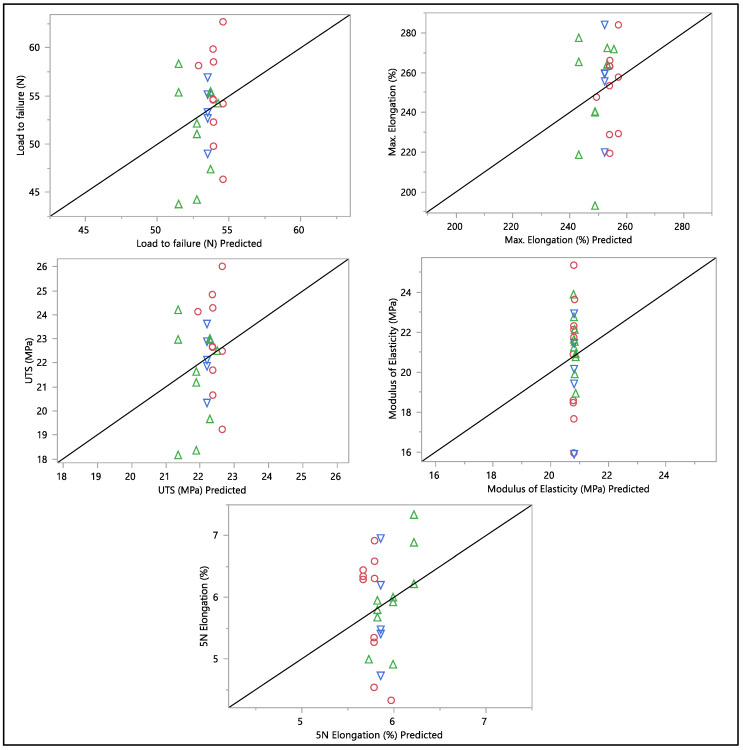
Actual vs. predicted plots of load to failure, percentage of elongation to failure, 5 N percentage of elongation, ultimate tensile strength, and modulus of elasticity. Different colors mean different in-vivo duration exposure.

**Figure 12 bioengineering-11-00156-f012:**
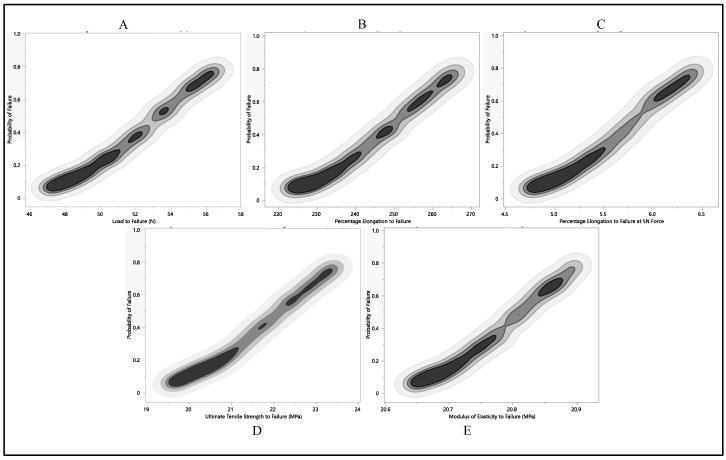
Probability of failure to each residual property with respect to Monte Carlo simulation, (**A**) load to failure, (**B**) percentage of elongation to failure, (**C**) percentage of elongation at 5 N force, (**D**) ultimate tensile strength, (**E**) and modulus of elasticity.

**Table 1 bioengineering-11-00156-t001:** Residual properties of the tested left ventricle polyurethane leads (n = 10; 2–3 test samples from each of the leads).

In Vivo Duration in Months	Load to Failure (N)	Max. Elongation (%)	5 N Elongation (%)	UTS (MPa)	Modulus of Elasticity (MPa)
<1	46.35	229.30	6.34	19.23	20.90
<1	54.19	257.70	6.44	22.49	18.47
<1	62.69	283.96	6.29	26.01	18.59
13	54.22	271.64	4.99	22.50	23.89
13	53.73	251.78	4.95	22.29	26.49
24	49.78	263.07	5.27	20.66	21.75
24	58.53	266.08	5.35	24.29	22.32
24	52.28	219.39	4.54	21.69	25.36
25	54.66	253.37	6.59	22.68	17.66
25	54.59	228.80	6.92	22.65	15.95
25	59.86	263.21	6.31	24.84	22.14
31	47.37	264.15	5.67	19.66	21.24
31	55.44	263.89	5.95	23.00	22.75
31	55.33	272.21	5.79	22.96	21.64
38	53.32	284.07	6.96	22.12	21.46
38	55.16	259.28	6.20	22.89	20.16
38	52.69	255.66	5.41	21.87	22.94
38	49.01	219.93	4.73	20.34	15.90
38	56.93	259.69	5.48	23.62	19.42
60	52.53	260.61	4.63	21.80	29.88
60	58.15	247.61	4.33	24.13	23.64
64	52.13	240.28	4.91	21.63	21.52
64	44.22	193.00	6.00	18.35	19.89
64	51.03	239.93	5.92	21.17	22.12
108	43.76	218.56	7.34	18.16	18.93
108	58.31	265.24	6.89	24.20	20.76
108	55.34	277.36	6.22	22.96	20.90

**Table 2 bioengineering-11-00156-t002:** Mathematical Predictive Modeling for Mechanical Properties in Materials Testing.

Prediction Model	Prediction Equation	Significancy with Respect to In Vivo Exposure	Behavior during In Vivo Exposure
Load to failure	54.6 − 0.029 **τ**	*p*-value = 0.82	Decrease
Percentage of elongation	257.22 − 0.1302 **τ**	*p*-value = 0.62	Decrease
Percentage of elongation at 5 N	5.66 + 0.005 **τ**	*p*-value = 0.0066	Increase
Ultimate tensile strength	22.63 − 0.034 **τ**	*p*-value = 0.82	Decrease
Modulus of elasticity	20.8 − 0.0007 **τ**	*p*-value = 0.12	Decrease

## Data Availability

Data are contained within the article.

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
