# Peer review of "Mechanical Behavior of Polyurethane Insulation of CRT Leads in Cardiac Implantable Electronic Devices: A Comparative Analysis of In Vivo Exposure and Residual Properties"

_bioengineering, 2024, doi:10.3390/bioengineering11020156_

Round 1
Reviewer 1 Report
Comments and Suggestions for Authors
The study explored the mechanical performance of polyurethane leads after explanation. They compared the durability after difference times of implanation and found there were no signficance. However there are some questions raied.
1. The author mentioned poluurethane as insulation material and aslo SI-polymide as inner insulation materials. Could the authors explained more the desgined for insulation, and lubricity and mechanical strenghth based on these materials in more details?
2. The author should mentioned the manufactuerer's name since it is a comemercialized product.
3. The author exam the integrity of leads after explanation. However, the sensing, stimulation properties should also be exam after explanation as comparison to new product.
4. The author should provide the usual condition in human or animal data. (such as the propotion of force usually faced in CRT device)
5. The duration of implanation was not long, since CRT lead was designed for implantation for a patient's life time, except for one lead, which was 108 month. However the longest implanted lead had significant difference in lead perfomance (such as load to failure). Is this the data adequate to sustain pacing and sensing in human?
Comments on the Quality of English LanguageEnglish is fine
Author Response
Dear Reviewer,
The authors are grateful for your comments and appreciate your effort in trying to improve this article. Please find below the response to the comments you made. We have addressed each of the comments in the manuscript as well as in the attached file.
Please see the attachement

Reviewer 2 Report
Comments and Suggestions for Authors
Your paper needs a series of clarification in the methods section as highlighted in the reviewed paper. In addition your statistical methods should be outlined as well as how you established and verified your prediction model.
Check references for relevance to your topic.
Improve figure 1 d in order that the reader can see the details of fixing the lead into the brackets/holder of your test machine. Indicate the model used and the accuracy provided by that particular equipment.
It is unclear how you managed that you were able to measure the properties of the outer PUR insulation layer.
A series of questions remain as highlighted in the pdf file

The authors describe the results of mechanical properties of three different types of Medtronic pacemaker leads. They have obtained these leads from patients (or cadavers) after explantation with duration of use between 1 and 108 months. The authors already have published at least three similar investigations with other types of pacemaker leads which they refer to in the references.
There is the urgent need of clarifications throughout their paper as highlighted in my review
Author Response

(The authors gave the same response as above.)

Reviewer 3 Report
Comments and Suggestions for Authors
I thank the authors for giving me the privilige to comment on their paper. As I am a surgeon and not a bioengineer I am sorry that I might not give sufficient feedback to the manuscript.
One question which came into my mind was what the exact aim of this study is. As a clinician I assume that these kind of teste have been done by the manufactor? It would,be helpful if you could describe the aim of the study in more detail so that it is understandable for clinicians.
Second, what are the excat consequences of the study? I am not sure if clinicians understand the conclusion of the study.
Further, is this data forwarded to the FDA?
Thank you again for considering my questions.
Author Response
Dear Reviewer,
The authors are grateful for your comments and appreciate your effort in improving this article.
Please see the attachment.

Round 2
Reviewer 1 Report
Comments and Suggestions for Authors
The authors answered my questions well. Their work merits publication in Bioengineering
Author Response
Dear Reviewer,
Thank you for your time and effort.

Reviewer 2 Report
Comments and Suggestions for Authors
Your revision and your answer is appreciated. Anyway please strengthen you paper by considering my comments as suggested in the comments in the revised paper. Find the suggestions in the sticky notes.
iyour answer item 5: from your comment given here it seems that you wanted to test the PU outer insulation, If this is the case it needs a better clarification in your revised paper. The inside coils insulated with SI-PI will stretch within their mechanical properties with low or even no effect on the PU insulation - correct? Inform the reader about this assumption!
However, if you think you single out the properties of the PU insulation, then the ultimate strength calculation of the PU insulation hardly can be based on the surface area of the lead.
The explanation given here should be added to your text, eventually in a short appendix precisely explaining your test set up. Then also your term 5N elongation failure.
item 22: as you could show in your investigation the "degradation of material" is minimal and therefore a lead malfunction is not to be expected from this kind of changing material properties. Of couese a broken coil or displacement of the stimulating electrode could endanger the patient
item 27, 28, 30: deliver the message to the reader

fine, check the word seperations
Author Response
Dear Reviewer,
The authors are grateful for your comments and appreciate your effort in trying to improve this article. Please find below the response to the comments you made. We have addressed each of the comments in the manuscript in the attached word file.
Thank you,
